# Skilful forecasting of global fire activity using seasonal climate predictions

Marco Turco [1], Sonia Jerez[2], Francisco J. Doblas-Reyes[3,4], Amir AghaKouchak [5], Maria Carmen Llasat [1] & Antonello Provenzale[6]

Societal exposure to large fires has been increasing in recent years. Estimating the expected fire activity a few months in advance would allow reducing environmental and socio-economic impacts through short-term adaptation and response to climate variability and change. However, seasonal prediction of climate-driven fires is still in its infancy. Here, we discuss a strategy for seasonally forecasting burned area anomalies linking seasonal climate predictions with parsimonious empirical climate–fire models using the standardized precipitation index as the climate predictor for burned area. Assuming near-perfect climate predictions, we obtained skilful predictions of fire activity over a substantial portion of the global burnable area (~60%). Using currently available operational seasonal climate predictions, the skill of fire seasonal forecasts remains high and significant in a large fraction of the burnable area (~40%). These findings reveal an untapped and useful burned area predictive ability using seasonal climate forecasts, which can play a crucial role in fire management strategies and minimise the impact of adverse climate conditions.

[1] Department of Applied Physics, University of Barcelona, 08028 Barcelona, Spain. [2] Regional Atmospheric Modeling Group, Department of Physics, University of Murcia, 30100 Murcia, Spain. [3] Barcelona Supercomputing Center (BSC), Carrer de Jordi Girona 29-31, 08034 Barcelona, Spain. [4] ICREA, Pg. Luís Companys 23, 08010 Barcelona, Spain. [5] Department of Civil and Environmental Engineering, Center for Hydrometeorology and Remote Sensing, University of California, Irvine, CA 92697, USA. [6] Institute of Geosciences and Earth Resources (IGG), National Research Council (CNR), 56124 Pisa, Italy. Correspondence and requests for materials should be addressed to M.T. (email: turco.mrc@gmail.com)

O
ver the past 30 years, the development of seasonal climate prediction models has grown from pure research to routine operational activities[1] across a range of applications around the world (e.g. energy and water management, insurance, agriculture[2,3]). However, studies assessing the skill of seasonal climate predictions (as obtained from dynamical climate models) to forecast fire burned areas (BA) are still relatively scarce[4–8] and mostly limited to a single season or region. Moreover, most studies that exploit the use of statistical models for forecasting fire activity based on climate information rely on few predictors and have regional focus[9–11]. Lack of long-term global fire data, needed to establish solid empirical or statistical relationships between climate and fire activity as the basis to predict BA, has prevented global scale studies[12]. The situation has recently changed as the global dataset of monthly BA described in Giglio et al.[13] is now spanning the last two decades, making comprehensive analysis of the climate–fire links worldwide possible[14,15]. However, a global assessment of dynamical seasonal climate forecast systems to be used for BA prediction has not been addressed so far.

The overarching goals of this study are to develop empirical predictive relationships between fire and climate variables for the entire globe and to explore the performance of an integrated climate–BA model that combines empirical fire–climate models with global climate seasonal forecasts, to obtain seasonal predictions of fire activity worldwide.

The key contribution of this study is to assess the current skill of BA predictions using multi-model seasonal climate predictions at a global scale and for each season separately. The results revealed substantial BA predictability based on antecedent and forecasted climate conditions that can be exploited for fire risk management months ahead. Our study could serve as the basis for the development of a global fire seasonal forecast product.

## Results

**Defining the climate–fire model with observations.** Precipitation is a first-order driver of BA globally[16]. For this reason, and after evaluation of other potential climatic drivers, here we selected the standardized precipitation index (SPI[17,18]) as the climate indicator/predictor of BA. SPI transforms accumulated precipitation values over a specific period (usually from 1 to

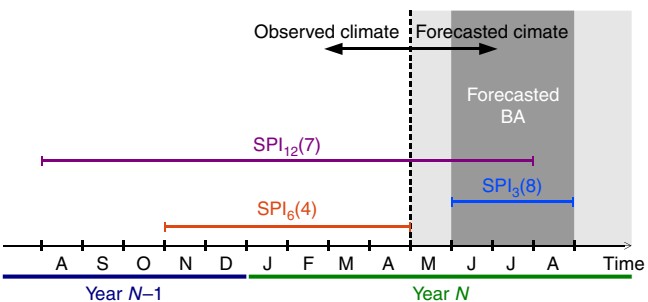

**Fig. 1** Schematic view of the proposed burned area forecast system. In order to forecast burned area (BA) in JJA of year $N$ (dark grey shadow), we rely on the climate forecast issued on May of the same year (dashed line). Before May, we have the observed climate data, while from May on (light grey) we only have seasonal climate forecasts. Observations and forecasts can be merged to construct the SPI$_t(M-m)$−BA model, depending on the values of the parameters $t$ (which can take a value of 3, 6 or 12 months) and $M-m$ (which, in the illustrated example, can vary between March and August of year $N$, i.e. from the last month of the season being forecasted, back to the prior 6 months). As examples, we represent how climate observations and forecasts should be merged to compute SPI$_{12}(7)$ (purple line); how SPI$_3(8)$ is constructed from climate forecasts only (light blue line); and how SPI$_6(4)$ (orange line) is computed only from observations

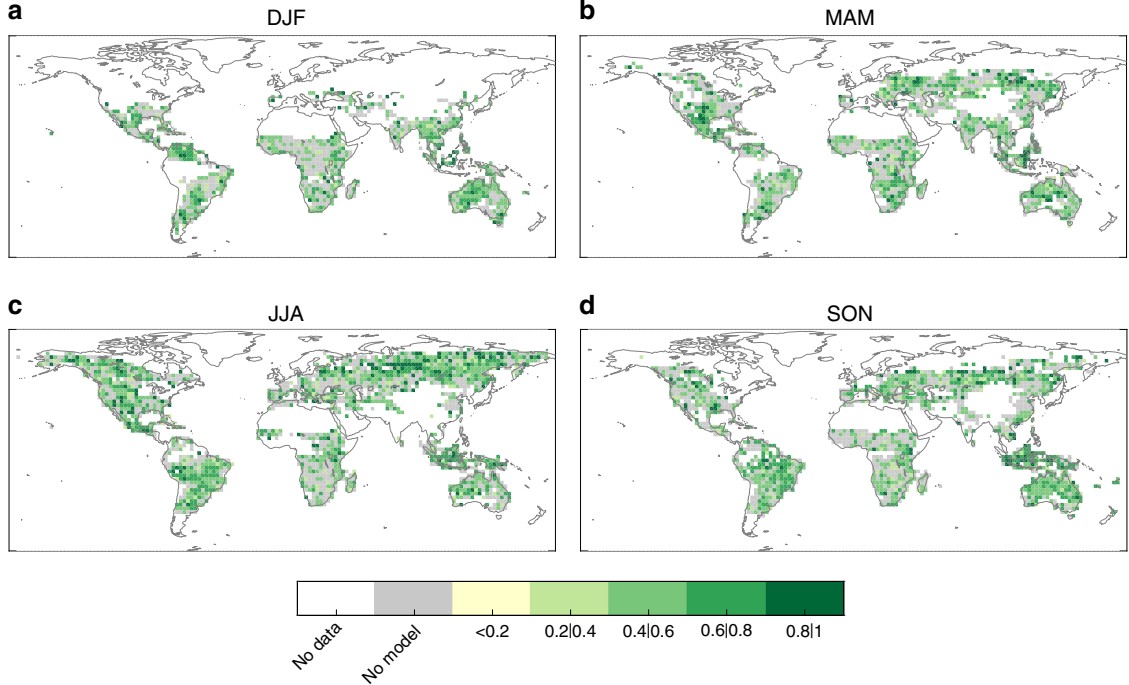

**Fig. 2** Maximum skill of burned area predictions obtained using observed climate. Correlations of out-of-sample burned area (BA) predictions using the SPI–BA model fed with observed SPI data for **a** December–January–February (DJF), **b** March–April–May (MAM), **c** June–July–August (JJA) and **d** September–October–November (SON). Only correlations that are significant ($p$-value < 0.05) are shown in green colours. Grey colour shadows those grid-points with non-significant correlation values. White indicates areas where fires do not occur (e.g. sea) or have not been recorded

12 months) into a standard Gaussian distribution with zero mean and unit variance, with positive and negative values indicating wet and dry conditions, respectively. In addition to SPI, we also explored other indicators and variables including the standardized precipitation evapotranspiration index (SPEI[19]), temperature, and a regression-based precipitation–temperature indicator (i.e. a linear combination of SPI and temperature).

For each point of the global grid (at a $2.5° \times 2.5°$ spatial discretization) and for each season separately (December–January–February, DJF; March–April–May, MAM; June–July–August, JJA; September–October–November, SON), we express the possible link of year-to-year changes in BA with the SPI (and other indicators) using the following model:

$$BA = \beta \cdot SPI_t(M - m) + \varepsilon \qquad (1)$$

In Eq. (1), $\beta$ represents the sensitivity of BA to dry or wet conditions as informed by the SPI, $m$ is the month for which the SPI is computed (which we allow to vary from $M-5$ to $M$, where $M$ is the last month of the season considered; see Fig. 1), $t$ is the accumulation time window (number of months) used to compute the SPI (we consider periods of 3, 6 and 12 months; for instance, $t = 3$ thus corresponds to precipitation anomalies accumulated over the three months $m-2$, $m-1$ and $m$; see Fig. 1), and $\varepsilon$ is a stochastic noise term that captures all other (neglected) factors that influence BA other than SPI. With this approach, we take into account the potential effect of antecedent climate conditions on BA, as described in previous works[20,21]. Prior to the analysis, the time series of fire and SPI data were linearly detrended (to minimise the influence of slowly changing factors; see e.g. Andela et al.[16] and Turco et al.[22,23]) and standardized (see Methods).

First, we determine empirically the best SPI–BA model for each grid point and season. The approach is based on finding the values of the model parameters ($\beta$, $m$ and $t$) that maximise the correlation ($r$) between modelled and observed BA series. We assess the performance of the model to achieve out-of-sample BA predictions from the knowledge of the predictor SPI data outside the period used to train the model, adopting a leave-one-out cross-validation method. In the model, we use the observed SPI values for the 21-year long period for which the BA series are available (see Methods).

Figure 2 shows the correlations between the out-of-sample BA predictions, obtained using the observed SPI data as drivers, and the observed BA series. These results provide the maximum skill of BA using the SPI–BA model as they are obtained using the best available climate data (that is, observational references) as drivers. We find that in a substantial fraction of the domain area (about 60% depending on the season) such correlations are statistically significant, with an average correlation value, $\langle r \rangle$, of 0.57–0.59 depending o n the season. There is thus a promising basis for developing a seasonal fire forecast system based on operational dynamical climate forecast systems, as illustrated below.

To support our choice of SPI as the best predictor for BA, we show a comparison of the BA predictions using other indicators and variables including SPEI (which is mathematically similar to SPI, but including also potential evaporation), temperature alone (T model) and a multiple liner regression-based model using temperature and SPI (SPI-T model; i.e. BA $= \beta \cdot SPI_t(M-m) + \gamma \cdot T_t(M-m) + \varepsilon$). Figure 3 summarises the results for all seasons, models and verification metrics. The T model shows the worst performance, because only 40–45% of the domain area has significant correlations (with values on the order of 0.5–0.55; Fig. 3a). The SPEI, SPI and SPI-T models perform very similarly, with around 60–65% of the global burnable area showing statistically significant correlations between modelled and observed BA series (with an average correlation value of around 0.55; Fig. 3a). Similar conclusions are drawn using the

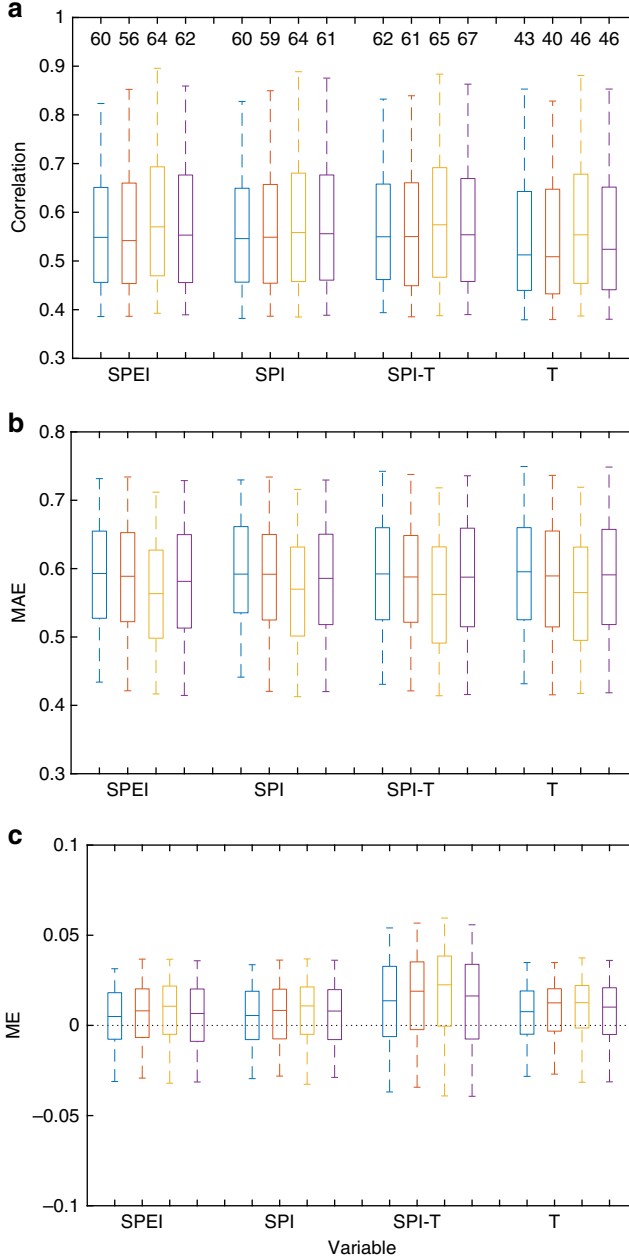

**Fig. 3** Summary of burned area prediction skill obtained using different observed climate indicators and metrics. Boxplots of the spatial distribution of **a** correlation values (numbers above the boxes represent the percentage of the domain area with significant correlations, i.e. number of green against grey plus green points in Fig. 2), **b** mean absolute error values (MAE) and **c** mean error (ME) for the burned area predictions based on SPEI, SPI, SPI and temperature (SPI-T model) and temperature alone (T model) for the four seasons (depicted with different colours). The median is shown as a solid line, the box indicates the 25–75 percentile range while the whiskers show the 2.5–97.5 percentile range

mean absolute error (MAE) metric (Fig. 3b). These results confirm that precipitation alone explains much of the year-to-year BA variability on a global scale[16]. Finally, the mean error (ME) metric, that measures the difference between the average prediction and observation, indicates that the systematic error is low, with values between −0.05 and 0.05 (in standard deviation units; Fig. 3c), with the largest range of values

**Table 1 Seasonal forecast systems considered in this study**

| Model acronym | Description | References |
|---|---|---|
| ecmwf-s4 | ECMWF Seasonal Forecast System 4 | Molteni et al.[58] |
| ecmwf-s5 | The fifth generation of the ECMWF seasonal forecasting system | Available user guide at https://www.ecmwf.int/sites/default/files/medialibrary/2017-10/System5_guide.pdf (accessed March 14, 2018) |
| cfs-v2 | NCEP coupled forecast system model version 2 | Saha et al.[59] |
| cancm4 | Canadian Centre for Climate Modeling and Analysis Coupled Climate Model version 4 | Merryfield et al.[60] |
| cm2p5-flor-a06 | Geophysical Fluid Dynamics Laboratory Climate Model version 2.5, flor version a06 | Delworth et al.[61]; Vecchi et al.[62] |
| cm2p5-flor-b01 | Geophysical Fluid Dynamics Laboratory Climate Model version 2.5, flor version b01 | Delworth et al.[61]; Vecchi et al.[62] |
| rsmas-ccsm4 | The fourth version of the Community Climate System Model | Gent et al.[63] |

corresponding to the SPI-T model. Based on these analyses, the T and SPI-T models were discarded. While the performance of the SPI and SPEI models was very similar, the former has been selected allowing for defining a very simple and parsimonious climate–BA model.

**Feeding the climate–fire model with seasonal forecasts.** Here we assess the skill of retrospective forecasts (or re-forecasts) of BA, considering a lead-time of 1 month and using seasonal predictions as drivers. For instance, the forecasts of the total BA in JJA are obtained from the climate forecasts issued in May (see the illustration of Fig. 1), giving 1 month of lead time. We consider seven seasonal dynamical predictions (Table 1). Exploring the feasibility of BA predictions from operational multi-model products is an important novelty of this study.

Figure 4 shows the percentage of the domain with statistically significant correlation between predicted and observed BA, considering the different dynamical forecast systems, driver climate variables and seasons. For the sake of comparison, the results obtained with observations described above are also reported. Three main conclusions can be drawn from this analysis. First, we further confirm that the SPI is the best predictor when the BA prediction model is fed with actual seasonal forecasts (note that Figs. 2 and 3 were obtained using climate observations to feed the BA prediction model) after balancing performance and parsimony among the various approaches. Second, quite similar results across seasons were found. Third, among the various seasonal forecast products, the best results are achieved with the seasonal forecast systems cfs-v2, ecmwf-s4 and ecmwf-s5.

We also explored how to best combine the various forecasts products to obtain the most skilful predictions, as ensemble means of multiple forecast models typically have better skill than any particular model[24–28]. We considered two different ensembles: the ensemble mean of the seven forecast systems (ENS hereinafter), and the ensemble mean of the three best performing models (BESTENS hereinafter, i.e. cfs-v2, ecmwf-s4 and ecmwf-s5). The two lower rows in Fig. 4 show that ENS reaches a similar percentage of global burnable area with skilful BA predictions as the best single model, while BESTENS systematically outperforms the individual models. Consequently, we consider the ensemble BESTENS results in the following.

Figure 5 shows the correlations between the predicted and observed BA series in each season, using the SPI predicted from the BESTENS seasonal climate forecasts for the BA predictions. These results allow for determining the skill of our forecast system to produce BA predictions. Whilst the predictive capability of the model is reduced when compared to the results of Fig. 2, the skill is still high ($\langle r \rangle$ from 0.55 to 0.57 depending on the season) and significant over a large fraction of the

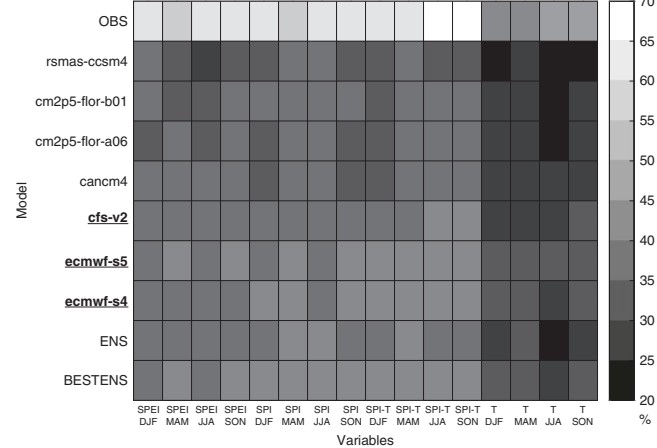

**Fig. 4** Percentage of global burnable area with skilful burned area predictions obtained from various seasonal forecast systems. Different rows indicate different climate forecast systems (labelled according to Table 1), including the burned area (BA) predictions obtained with observation (OBS), with the ensemble mean of all the models (ENS), and the ensemble mean of the best models (BESTENS; the best models are highlighted with underlined and bolded name). Different columns indicate the performance of the BA prediction model based on different predictors (SPEI, SPI, SPI and T, and T) for the different seasons

domain (about 40% depending on the season). The regions where significant correlations are found include also extra-tropical areas, such as Mediterranean Europe and the central-northern Asian regions, where dynamical forecast systems are known to have a limited prediction skill[1,29,30]. The skill found here largely relies on merging observational information (for the months previous to the fire season) with seasonal forecasts (for the fire season). The MAE of the BESTENS is slightly higher than the BA prediction using observations, as expected, and the ME is between −0.15 and 0.1 (Fig. 6). To complete the BA skill assessment, we also evaluated the added value of the forecast model framework against a null model obtained by considering only long-term averages of observed BA (i.e. a forecast based on BA climatology). Figure 7 confirms that the forecast model produces higher correlations than the null model, supporting the usefulness of current seasonal forecast systems over a naïve climatology estimate.

Clearly, an improvement of seasonal predictions would further enhance the usefulness of the SPI–BA model discussed here. In this sense, the above results are conditioned on the skill of the SPI (see Supplementary Figures 1–4) and on the characteristics of the SPI–BA model. The values of the parameters $\beta$, $m$ and $t$ leading to

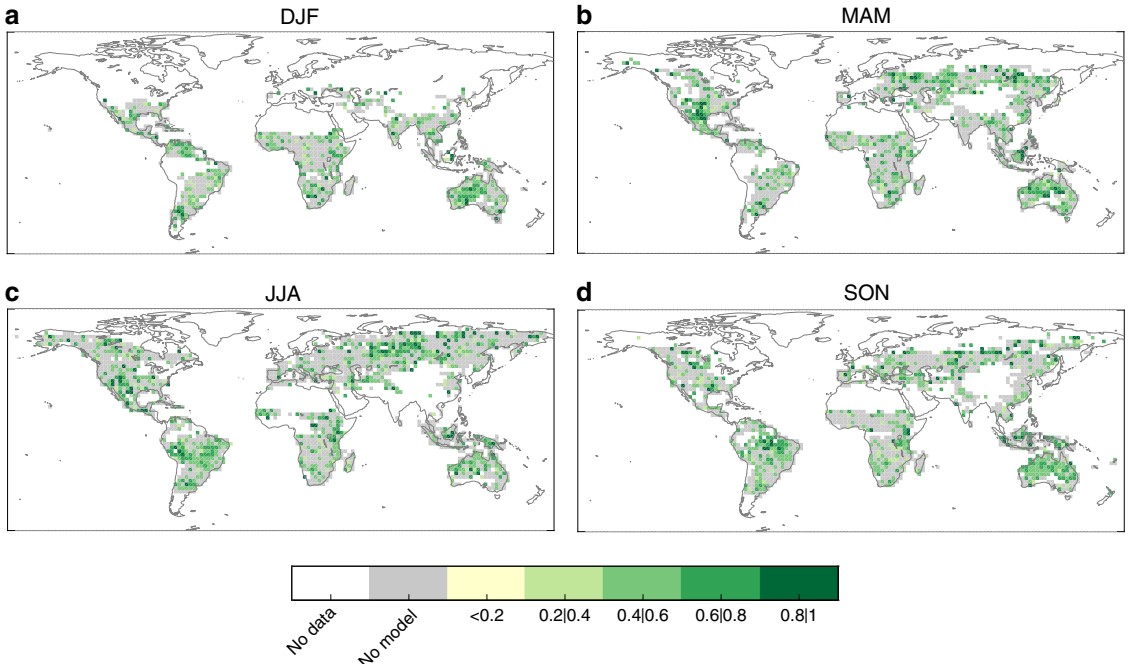

**Fig. 5** Skill of burned area predictions obtained from the seasonal forecast ensemble BESTENS. Correlations of out-of-sample burned area (BA) predictions using the SPI–BA model fed with seasonal forecasts of SPI from the BESTENS for **a** December–January–February (DJF), **b** March–April–May (MAM), **c** June–July–August (JJA) and **d** September–October–November (SON). Only correlations that are significant (*p*-value < 0.05) are shown in green colours. Grey colour shadows the grid points with non-significant correlation values. White indicates areas where fires do not occur (e.g. sea) or have not been recorded

the correlations shown in Figs. 2 and 5 are displayed in Fig. 8a–l. This information provides insights on the way climate affects fire activity and on the sources of BA predictability worldwide. The coefficient $\beta$ (the average across all the out-of-sample estimates; Fig. 8a, d, g, j), representing the response of BA to SPI variations (that is, the fingerprint of climate on BA), is generally negative (73%, 72%, 78% and 72% of the domain with SPI–BA model, for DJF, MAM, JJA and SON, respectively). Since negative SPI values correspond to dry conditions, this intuitively indicates that in most regions drier conditions led to larger BA values. These results agree with other studies that focused on regions with abundant fuel but rarely dry ecosystems, where fires are mainly limited by fuel moisture, generally indicating that drier conditions promote larger fire activity[21,31]. Nonetheless, there are also areas (for instance Australia and central-eastern Africa) where the BA sensitivity to SPI is positive ($\beta > 0$). These are arid regions, where fire spread is mostly limited by the fuel amount, which is enhanced by antecedent wet conditions[32]. We also found that the spatial correlation between the climatological patterns of annual water balance (provided in Supplementary Figure 5) and the spatial pattern of $\beta$ is statistically significant and negative, with correlation values of −0.41, −0.29, −0.27 and -0.40 respectively in DJF, MAM, JJA and SON. This suggests that droughts play a prominent role in wetter areas, while wet conditions can promote larger fires in arid regions. These results are in line with the intermediate fire–productivity hypothesis[20,33], which suggests that fire activity reaches two minima, one dominated by high aridity values where fire spread is mostly limited by fuel amount, and another characterised by low aridity where fuels are abundant and fires are mainly limited by fuel moisture content. In regions with large climate/ecosystem gradients, substantially different climate–fire links can exists close to each other. In such areas, we should acknowledge that the spatial resolution of our analysis

might obscure the relationship, hampering local/regional interpretations.

Figure 8 (panels b, c, e, f, h, i, k, l) also shows the time scales (i.e. duration of dry/wet periods) and timing of climate conditions that more strongly influence fires across the globe (parameters $t$ and $m$, respectively). Although the spatial variability of these patterns is quite high, some distinct behaviour can be inferred, in keeping with the discussion above. Overall, short-term drought conditions (concomitant with the fire season) lead to larger BA in humid regions (e.g. northern Asia in JJA), while antecedent wetter conditions over longer accumulation periods favour higher values of BA in arid areas (e.g. Australia). The values of $m$ are generally close to the end of the fire season considered (Fig. 8b, e, h, k). For instance, over extended regions, in the optimisation of the SPI–BA model the SPI of February/August is selected for the DJF/JJA season. In these cases it is necessary to resort to the four-months-ahead predictions of precipitation to compute the SPI. There are also regions (e.g. Australia) where BA is related to antecedent SPI in such a way that dynamical climate forecasts are unnecessary (i.e. $m \leq 4$; prior to November/May for forecasting DJF/JJA BA). Clearly, where antecedent observed SPI allows predicting BA months in advance, the SPI seasonal forecast errors do not affect the skill of the BA prediction model. Also, the SPI and thereby BA prediction skill increases with larger values of $t$ (Fig. 8c, f, i, l), since our forecast employs more observed data for longer SPI accumulation windows[34]. In fact, merging observational information (for the months previous to the fire season) with seasonal forecasts (for the fire season) is a special feature of our approach that substantially contributes to increase fire predictability, making the most of the best information available to the users. This is especially useful over areas where the performance of the dynamical forecast systems is still affected by significant errors. For instance, our models show

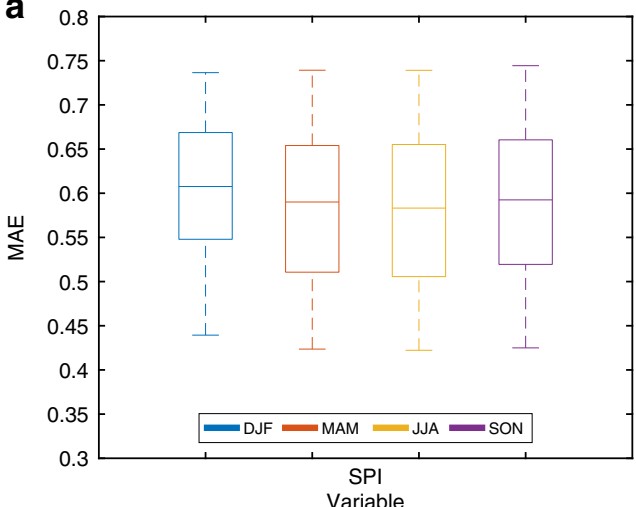

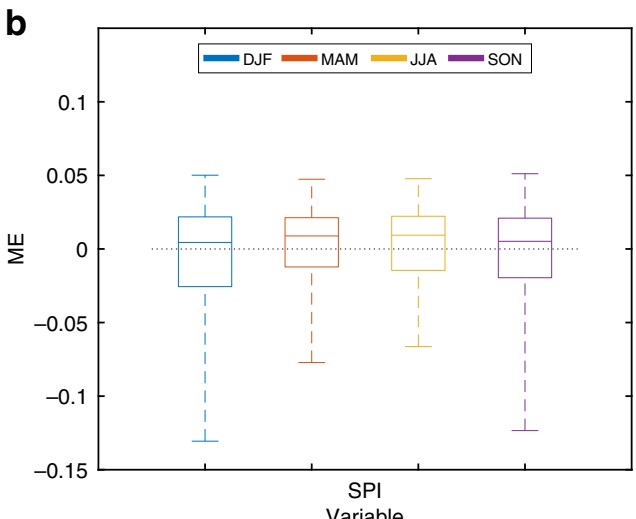

**Fig. 6** Verification summary of burned area predictions obtained from the seasonal forecast ensemble BESTENS. Boxplots of the spatial distribution of the **a** mean absolute error values (MAE) and **b** mean error values (ME) for the burned area predictions using seasonal forecasts of SPI based on the BESTENS for the four seasons (depicted with different colours). The median is shown as a solid line, the box indicates the 25–75 percentile range while the whiskers show the 2.5–97.5 percentile range

skill also in mid-latitude regions, where dynamical forecast systems show acceptable skill only for particular seasons and events (see e.g. Frías et al.[35]; Doblas-Reyes et al.[1]; Ceglar et al.[36]).

## Discussion

Predicting fires is a challenging issue owing to the complexity of the processes involved, limitations in observational data and concurrence and compounding effects of multiple drivers. Bearing this in mind, we proposed a parsimonious mathematical model to describe the impact of climate variability on BA. Assuming climatic processes act as top-down controls on the regional pattern of year-to-year changes in fire, we provided seasonal BA predictions. Our study provides a basis for the development of a global fire seasonal forecast product. In this context, it is worth noting that the generalisation of the proposed method is technically straightforward. For applying our approach

to continuously-updated fire forecasts to cover all trimesters of the year, one should resort to seasonal forecasts issued every month for rolling three-month periods (e.g. JFM, FMA,…, DJF). The development of a prototype real-time operational forecast system, however, may be challenging owing to the uncertainties of the observed near-real-time data, especially over data-poor regions such as Africa and South America[37,38]. Thus, although actionable near-real datasets are available (see e.g. Janowiak and Xie[39]; Chen et al.[40]), it is recommended that, before implementing our approach for real-time application, a careful assessment of the available data sets is performed. This system is not designed to replace existing systems that are currently in use. Instead, it offers complementary information to the existing systems while providing a global perspective. Some of the key drivers of fires (e.g. droughts, high temperatures) often affect extensive areas beyond national boundaries. For this reason, extreme fires can affect multiple countries, justifying the efforts for a transnational system for fire prediction and risk management. The proposed modelling framework offers a unique avenue to move toward such a system.

This first assessment of seasonal prediction of BA on a global scale, based on dynamical seasonal climate forecasts represents a baseline study for future analyses. Possible future developments include more refined fire-specific seasonal climate forecast systems, improved climate–fire data products, more sophisticated empirical methods with better calibration of the predictors, other climatic variables (see e.g. Williams et al.[41] that consider the Vapor Pressure Deficit, or Turco et al.[42] that consider the standardized soil moisture index), use of probabilistic forecasts, and/or higher spatial resolution. In particular, given the still rather moderate skill of seasonal forecasts, further efforts are clearly necessary to increase the forecast quality of the climate conditions. Also, as the length of the records and the quality of global fire datasets increase over time, climate–BA models may become more accurate. Despite current limitations in observations and model predictive skill, the results reported here contribute to a better characterisation of the climate–fire relationship. We show that in most regions the BA is inversely associated with SPI (negative correlation). Given that negative SPI values correspond to dry conditions, this suggests that, as expected, drier conditions lead to larger BA values. In a changing climate, several possible pathways of fire response can be identified – depending on the expected changes in precipitation, temperature, vegetation and human activities[43–45]. With respect to the direct impact of climate change in regulating fuel moisture (i.e. prolonged droughts and warmer climate leading to larger fires), fire risk is expected to increase where the climate is projected to become warmer and drier[46,47]. Despite long cohabitation of humans and fires[48], our fire management abilities and response still remain limited in most part of the world[49,50]. We hope that the proposed BA forecasting model evolves into a long-term predictive system that can be used in decision-making and operational applications.

## Methods

**Climate and fire data**. We consider three climate indices/variables: SPI[17,18], SPEI[19] and air temperature (T). SPI is a transformation of the accumulated precipitation values over a specific period (here over 3, 6 and 12 months) into a standard Gaussian distribution with mean 0 and standard deviation 1. Positive values indicate surplus of rainfall, whereas negative values identify dry conditions relative to the long-term climatology. The SPEI is mathematically similar to SPI. It estimates the monthly water balance as precipitation minus potential evapo-transpiration (e.g. estimated using the Thortnthwaite equation as in this study), and it is obtained through a standardisation of the multi-month (3, 6 or 12 months) water balance values. For SPI and SPEI the standardisation step is based on a nonparametric approach in which the probability distributions of the data samples are empirically estimated[51,52]. We calculate the T indicator as multi-month averages (over 3, 6 and 12 months) of monthly temperature data and then we obtain standardized series by (a) defining an anomaly by subtracting the long-

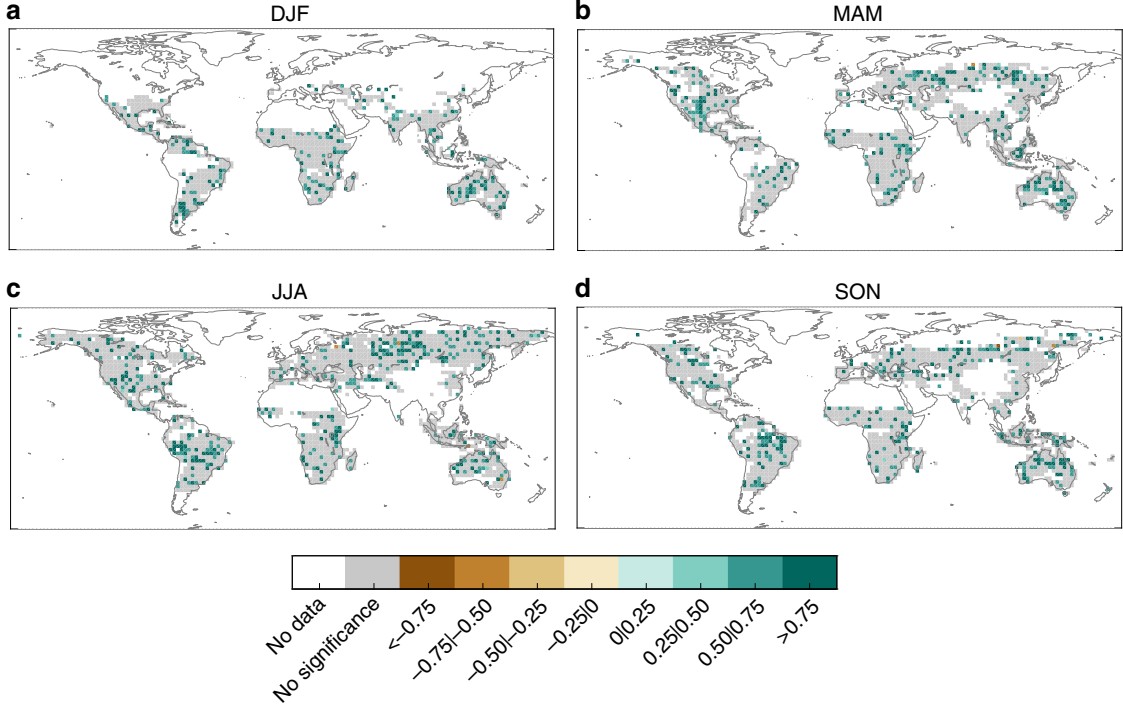

**Fig. 7** Comparison with predictions based on a null model. Differences in correlations of out-of-sample burned area (BA) predictions using seasonal forecasts of SPI based on the BESTENS ensemble and those based climatology, that is, considering only long-term averages of observed BA for **a** December–January–February (DJF), **b** March–April–May (MAM), **c** June–July–August (JJA) and **d** September–October–November (SON). Only significant differences are shown. Grey colour shadows the grid points with non-significant differences. White indicates areas where fires do not occur (e.g. sea) or have not been recorded

term mean from the original series and (b) dividing the anomaly by its long-term standard deviation.

We used two long-term and continuously-updated databases: ERA-Interim[53] for 2 m air temperature, with a resolution of 0.75° and GPCP[54] Version 2.3 for precipitation data, with a spatial resolution of 2.5°.

Seven seasonal models are used to provide temperature and precipitation forecasts (Table 1) including two from the European Seasonal to Interannual Prediction Project (EUROSIP[55]) and five from the North American Multimodel Ensemble (NMME[56]). These predictions are bias corrected by means of simple linear scaling performed by using a leave-one-out cross-validation, i.e. excluding the forecasted year when computing the scaling parameters. Specifically, we bias corrected the model ensemble mean at each grid-point considering, for precipitation, a scaling factor based on the ratio of long-term monthly means (over the period 1981-2016) of the observed and simulated data, while for temperature, we consider the difference of long-term monthly means of the observed and simulated data to correct the raw data (see Turco et al.[34] for more details). This scaling factor is lead-time and starting-date dependent, thus varying for each forecasted month and for each issued forecast. This bias correction aims at avoiding possible inconsistencies between simulated and observed data when both are merged to construct the predictors.

Monthly BA data were obtained from the GFED4[13] dataset for the period 1995/06–2016/05 with a spatial resolution of 0.25°.

To ensure consistency among the spatial resolution of the different datasets, and account for missing data in the BA time series, all the datasets are remapped from their original resolution onto the coarsest grid, defined by GPCP (2.5° × 2.5°). BA remapping involves summing up all the grid-points at 0.25° that fall in a 2.5° grid-point. Also, following Chen et al.[15], we consider only those 2.5° grid-points where seasonal BA was non-zero in more than half of the available period (i.e. for each season we select those pixels where BA > 0 in at least 11 of the 21 seasons). The climate data are also remapped from their original resolution onto a 2.5° × 2.5° grid, with a bilinear interpolation for temperature and a first-order conservative remapping procedure for precipitation (using Climate Data Operators; https://code.mpimet.mpg.de/projects/cdo).

**Climate-fire model development**. The procedure to develop the empirical climate–BA model of Eq. (1) includes the following steps. First, the time series of BA and SPI (and similarly, SPEI and T) are linearly detrended to minimise the influence of slowly changing factors such as gradual increase in fire management and land-use changes. This ensures isolating the effects of climate anomalies on the year-to-year BA variability. It is worth noting that similar results have been

obtained with the original (i.e. without detrending) data (see Supplementary Figure 6). The SPI and BA anomalies are then normalised by subtracting the time-series mean and dividing by the standard deviation. This standardisation makes the regression results for the grid-points comparable with each other (i.e. we can compare across the domain the regression weights as they indicate how many standard deviations of BA anomalies change for every standard deviation unit change of the predictors). Then, to identify the best model we (i) fit all the possible versions of Eq. (1) considering all the potential predictors $SPI_t(M-m)$, with $t = (3,6,12)$, $m = (0,1,…, 5)$, $M$ is the last month of the season considered, i.e. same and previous fire-seasons months in agreement with previous studies (see e.g. Turco et al.[21]) through a leave-one-out-cross-validation; (ii) calculate the significance of the individual (Pearson) correlations of these models through a one-tailed hypothesis test; (iii) we seek the maximum correlation values among all the significant ($p$-value < 0.05) correlations calculated in the previous steps.

All the forecasts are done by using cross-validation in order to evaluate the predictions as if they were done operationally, including the steps of the bias correction of the seasonal climate data and of the calibration of the BA-climate models. Moreover, to avoid artificial skill, the observed series are de-trended and standardized in each step of the cross-validation, avoiding using observation of the predicted year. Both the linear trends and the regression coefficient of Eq. (1) are estimated using a robust regression procedure[57] that adopt iteratively reweighted least squares with a bisquare weighting function. Such an approach is less sensitive to outliers than the classic least-squares estimators.

**Code availability**. On behalf of reproducibility and applicability, the codes used in this work are available for research purposes by contacting the corresponding author. In any case the codes used for the data processing are mainly based on open source software: the Climate Data Operators (CDO version 1.7.2; functions: remapbil, remapcon) available from https://code.mpimet.mpg.de/projects/cdo, the netCDF Operator (NCO version 4.5.4; functions: ncwa -a ensemble) available from http://nco.sourceforge.net/, the R "Language and Environment for Statistical Computing" (R version 3.4.3, functions: thornthwaite from the R package SPEI, version 1.7) available from https://www.r-project.org/. The climate–fire model development and assessment is mainly based on Matlab codes written by M.T. that are available for research purposes from the corresponding author upon request.

**Data availability**. GFED4 data can be retrieved from the Global Fire Emissions Database (http://www.globalfiredata.org/data.html); GPCP Precipitation can be obtained from the NOAA/OAR/ESRL PSD, Boulder, Colorado, USA (https://www.

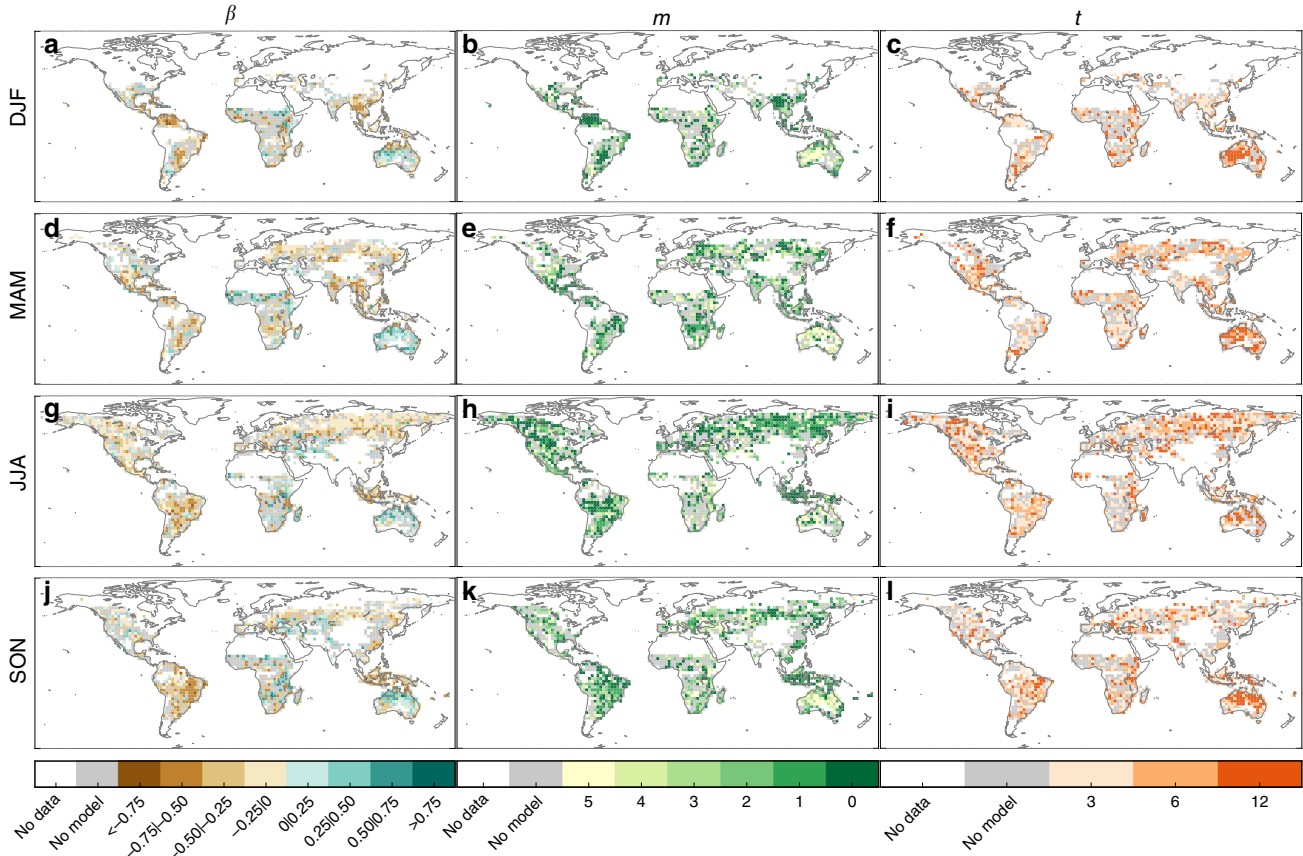

**Fig. 8** Spatial distribution of the parameters involved in the burned area prediction model. Spatial distribution of the optimal values of coefficient $\beta$ (**a**, **d**, **g**, **j**), of the lag ($m$) with reference to the last month $M$ of the season considered (**b**, **e**, **h**, **k**) and of the accumulation time scale ($t$: 3, 6 and 12 months; **c**, **f**, **i**, **l**) of the burned area-SPI model defined in Eq. (1) for the four seasons. Grey colour shadows the grid points with non-significant correlation values. White indicates areas where fires do not occur (e.g. sea) or have not been recorded

esrl.noaa.gov/psd/data/gridded/data.gpcp.html); ERA-Interim, ecmwf-s4, ecmwf-s5 data can be retrieved from the European Center for Medium Range Weather Forecast (https://www.ecmwf.int). The models cfs-v2, cancm4, cm2p5-flor-a06, cm2p5-flor-b01, rsmas-ccsm4 can be retrieved from the North American Multi-Model Ensemble website (http://www.cpc.ncep.noaa.gov/products/NMME/). In order to facilitate reproducibility and applicability of the proposal model, the authors will provide the data (observed and predicted) used in this study for research purposes to interested readers.

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

## Acknowledgements

This work was partially funded by the EU H2020 Project 641762 "ECOPOTENTIAL: Improving Future Ecosystem Benefits through Earth Observations" and the SERV-FORFIRE project of the ERA-NET for Climate Services, ERA4CS. M. Turco was supported by the Spanish Juan de la Cierva Programme (IJCI-2015-26953). F.J. Doblas-Reyes was supported by the H2020 IMPREX (GA 641811) and EUCP (GA 776613) projects. A.A. was partially supported by the National Oceanic and Atmospheric Administration (NOAA) award NA14OAR4310222, National Aeronautics and Space Administration (NASA) award NNX15AC27G, and National Science Foundation (NSF) INFEWS grant EAR 1639318. Special thanks to Esteve Canyameras and Xavier Castro for helpful discussions on the study.

## Author contributions

M.T. conceived the study. M.T. and S.J. designed and carried out the data analysis and wrote the paper. F.J.D.-R., A.A., M.C.L., and A.P. participated in defining the analysis and methodology, contributed to interpretation of the results, and to writing the paper.

## Additional information

**Competing interests:** The authors declare no competing interests.

