## [Peer Review File · Nature Communications]

Reviewers' comments:

Reviewer #1 (Remarks to the Author):

Building off previously published studies on the impact of drought on summer fires in Mediterranean Europe, the authors expanded the study to the whole globe at a 5 degree by 5 degree spatial resolution. Total area burned (BA) was optimized as a function of the cumulative standardized precipitation evapotranspiration index (SPEI) from the preceding months for each grid, using about 20 years of climate record and satellite-derived BA datasets. A strategy for seasonal fire activities forecasting with a 4 month lead time was then explored, based on similar BA-SPEI models but driven by seasonal climate forecasts from the European Centre for Medium-Range Weather Forecasts–System 4. The key findings include the relatively high and significant forecasting skill of BA, e.g., about half of the continental area, by just using preceding SPEI.

The paper addressed an important subject about the forecasting capability for seasonal wildfires using available operational seasonal climate predictions. The results of this paper and its implications are of general interest for audience, but they may not be in-depth enough to be compelling for the Nature Communications.

Major considerations:

1. The climate-BA models presented here are relatively simple - only one climate indicator (SPEI) was used across the globe, especially considering one individual model was constructed for each individual 5 degree grid. These models relied on a relatively short time series of data (about 20 years of record), and the robustness of the models is thus a bit limited, although the authors did their best to adopt a leave-one-out cross-validation method. A more systematic statistical approach would be more desirable, which considers a suite of potentially significant climate variables for seasonal BA predictions and automatically select the top significant variables for each grid or region based on the statistical analysis.
2. The prediction or forecasting skills were tested using the correlation as a single metrics. It would be more valuable to assess other statistics such as RMSE as well.
3. There are quite a few statistical climate-BA models for regional fire predictions, such as those in South America, continental US, and boreal regions. Some of them are at higher spatial resolution, and more likely have higher prediction accuracy by incorporating more predictors, and some use both spatial and temporal data for building climate-BA models that can explain both the spatial and temporal variability. From the stakeholders' perspective, it is not clear how this study, as presented, will further advance their knowledge and help them to improve fire management or climate adaptation strategies.
4. This paper presented a solid study in terms of testing the forecasting skills of using cumulative preceding SPEI, a drought indicator, to predict area burned at a global scale. It would likely fit a specialty journal better.

Reviewer #2 (Remarks to the Author):

Review of "Predicting climate-driven seasonal variations in wildfire activity worldwide"

The authors present an evaluation of the predictability and skill of global fires based on SPEI as estimated from observations and ECMWF S4 seasonal forecast. Although the topic is current and relevant, I would like to see some issues addressed before the paper is considered for publication in Nature Communications (detailed comments follow).

Main comments:

The authors argue this study could serve as the basis for the development of a global fire seasonal forecast product. Evaluating the seasonal forecast skill of one GCM is a first step, but I would be more convinced (for the purpose of effectively creating a fire forecast product) if the study was based on multiple forecast models, as ensemble means have typically better skill than any particular model alone (Robertson et al., 2004).

Robertson, A. W., Lall, U., Zebiak, S. E., & Goddard, L. (2004). Improved combination of multiple atmospheric GCM ensembles for seasonal prediction. *Monthly Weather Review*, 132(12), 2732-2744.

Line 168: The authors argue that their study serves as a baseline for future analyses, which will benefit from more developments, including a more refined seasonal climate forecast and higher spatial resolution. I suggest the authors revisit this argument as both the seasonal forecast model (ECMWF S4) and observational datasets are currently available at finer spatial resolution than that selected to derive the empirical relationships to BA (5 degrees). In addition, the coarser data set used in this study (GPCP at 2.5 degrees) can be replaced by other precipitation datasets namely CHIRPS, GPCC, CRU that are available at higher spatial resolution.

Line 177: The authors conclude that "... globally, the burned area is mainly influenced by concurrent drought and high temperature".

Mainly in what sense? If the model was averaging, globally, skill of $R=0.62$ in JJA that corresponds to less than 40% of BA variance explained. Please, clarify.

Also, the analysis presented here is based on SPEI and no evidence is presented to show that, for example, a BA model based on precipitation (or temperature) alone could have produced similar results. In other words, the sentence above suggests that among all the predictors, concurrent drought and high temperature are the most relevant, even though no other predictor is used.

Editorial Comments

Title: Burned area data does not distinguish whether fires are unplanned (wildfires) or prescribed/controlled fires, thus I suggest the authors explain how they defined wildfires or simply use the term fires.

Line 21: Fernandes et al. (2011) combines a statistical model with dynamical model seasonal forecast of SSTs to predict fires in the Amazon. It should be included in citation.

Fernandes, K., Baethgen, W., Bernardes, S., DeFries, R., DeWitt, D.G., Goddard, L., Lavado, W., Lee, D.E., Padoch, C., Pinedo-Vasquez, M. and Uriarte, M. (2011). North Tropical Atlantic influence on western Amazon fire season variability. *Geophysical Research Letters*, 38(12).

Line 54: What is a climatic balance? Do the authors mean water budget?

Line 56: Why such a coarse aggregation (5x5o) is used, given the datasets allow for a more refined grid?

Line 89: Since this is a global analysis, I suggest the season names (ex. summer) are replaced by the corresponding trimester.

Line 91: Lead time is defined as the period of time between the issue time of the forecast(May) and the beginning of the forecast period (June for JJA), thus in this case is a 1 month lead time, not 4.

Line 99: Do negligible ENSO impacts in these regions make the BA forecast more relevant since it

is able to overcome that limitation? If that is the case, why is that? Please, elaborate.

Line 160: Sentence: "... the performance of seasonal climate forecast system is still moderate,"
Add reference. Does this refer to specifically ECMWF S4? Is that true for both precipitation and temperature? Every season?

Line 227: Sentence: "For this reason, the time series of BA and of the SPEI index are linearly detrended to minimise the influence of slowly changing factors such as gradual increase in fire management effort and land-use changes".

I would like to point that observed trends in temperature have been shown to influence burned area anomalies regionally (Fernandes et al., 2017). Both observations and GCM hindcasts reproduce regional temperature trends and if the authors wish to place their findings in the context of climate change, as discussed in Lines 178-184, it would be worth to evaluate the sensitivity of BA forecast model to trends.

Fernandes, K., Verchot, L., Baethgen, W., Gutierrez-Velez, V., Pinedo-Vasquez, M., & Martius, C. (2017). Heightened fire probability in Indonesia in non-drought conditions: the effect of increasing temperatures. *Environmental Research Letters*, 12(5), 054002.

Other relevant references (consider citing)

Di Giuseppe, F., Pappenberger, F., Wetterhall, F., Krzeminski, B., Camia, A., Libertá, G., & San Miguel, J. (2016). The potential predictability of fire danger provided by numerical weather prediction. *Journal of Applied Meteorology and Climatology*, 55(11), 2469-2491.

Shawki, D., Field, R. D., Tippet, M. K., Saharjo, B. H., Albar, I., Atmoko, D., & Voulgarakis, A. (2017). Long-Lead Prediction of the 2015 Fire and Haze Episode in Indonesia. *Geophysical Research Letters*, 44(19), 9996.

Reviewer #3 (Remarks to the Author):

Comment on "Predicting climate-trivet seasonal variations in wildfire activity worldwide", by Turco and co-authors.

This study showed the usefulness of seasonal climate forecast in wildfire prediction. Near 50% of the continental areas benefit from significantly skillful forecast of wildfire based on seasonal climate model forecast. The analyses are valid, and physical interpretations of results are provided. The study is informative. I have a few comments here for consideration.

Comments:

1. The authors described the motivation and aims of the study. Can you also state what the novelty is of this study besides the spatial scale (global scale)?
2. When saying DJF and JJA seasons (also SON and MAM), for example in figure 2, do you mean the average of the 3 months in the season? i.e., DJF is the mean of Dec-Jan-Feb? If so, conventionally, I would call the forecasts of JJA initialized in May as lead-1-month forecasts rather than lead-4-month forecasts. Did you use data during 1981-2016 in figure 2?
3. To apply this to real wildfire forecasts, you may need calculations for every 3-month moving

window, e.g., JFM, FMA, DJF, provided that seasonal climate forecasts are initialized every month. So you will have forecast every month. But for research purpose, this is totally fine.

Responses to reviewer #1

Reviewer #1 (Highlight): *Building off previously published studies on the impact of drought on summer fires in Mediterranean Europe, the authors expanded the study to the whole globe at a 5 degree by 5 degree spatial resolution. Total area burned (BA) was optimized as a function of the cumulative standardized precipitation evapotranspiration index (SPEI) from the preceding months for each grid, using about 20 years of climate record and satellite-derived BA datasets. A strategy for seasonal fire activities forecasting with a 4 month lead time was then explored, based on similar BA-SPEI models but driven by seasonal climate forecasts from the European Centre for Medium-Range Weather Forecasts-System 4. The key findings include the relatively high and significant forecasting skill of BA, e.g., about half of the continental area, by just using preceding SPEI.*

The paper addressed an important subject about the forecasting capability for seasonal wildfires using available operational seasonal climate predictions. The results of this paper and its implications are of general interest for audience, but they may not be in-depth enough to be compelling for the Nature Communications.

Response: We wish to thank the anonymous referee for this review and his/her helpful advice. Each specific comment has been addressed in the manuscript and in this response document. Following the Reviewer's suggestion, we repeated all the analyses considering a higher spatial resolution, more models, other climate variables and we tested the skill with several statistical metrics. More details on the steps undertaken are now available in the revised paper. Most importantly, the results and conclusions of the previous version of our study are fully confirmed. Our findings reveal an untapped and useful BA predictive ability using seasonal climate forecast systems, which can play a crucial role in fire management strategies and minimise the impact of adverse climate conditions. Thus, given the relevance of the study to climate, fire and natural hazards communities, we believe that this study could be of general interest for the diverse readership of this journal.

Major considerations:

1. Referee's Comment: *The climate-BA models presented here are relatively simple - only one climate indicator (SPEI) was used across the globe, especially considering one individual model was constructed for each individual 5 degree grid. These models relied on a relatively short time series of data (about 20 years of record), and the robustness of the models is thus a bit limited, although the authors did their best to adopt a leave-one-out cross-validation method. A more systematic statistical approach would be more desirable, which considers a suite of potentially significant climate variables for seasonal BA predictions and automatically select the top significant variables for each grid or region based on the statistical analysis.*

Response: According to this comment and in response to a comment from another reviewer, we have redone all the analyses considering different potential predictors. In order to achieve this goal, we have considered four different model/indicators including: SPEI, as in the previous version of the manuscript; SPI (the standardized precipitation index); temperature (T) alone; and a multi regression model based on temperature and SPI (SPI-T). Results show that the models based on SPEI, SPI or SPI-T perform similarly, far better than T. Instead of using a different set of predictors for each grid point as the reviewer proposed, we prefer to use the same variable across the domain in order to ease the interpretation of the statistical model parameters. Thus, based on these analyses, we consider the SPI model as more robust since, even having similar performance, it is more parsimonious and therefore less susceptible to over-fitting. More details are available in the revised paper and in the Supplementary Information.

2. Referee's Comment: *The prediction or forecasting skills were tested using the correlation as a single metrics. It would be more valuable to assess other statistics such as RMSE as well.*

Response: We agree with the Reviewer. In the revised version of the manuscript, we verify the predictions considering the mean bias of the forecasts and the mean-absolute error (MAE), that measures the closeness of forecasts and observations. We prefer to use the MAE as it is considered a better indicator of average model performance than the RMSE (see e.g. Willmott & Matsuura, 2005), especially for short time series with skewed distributions that might have large values.

3. Referee's Comment: *There are quite a few statistical climate-BA models for regional fire predictions, such as those in South America, continental US, and boreal regions. Some of them are at higher spatial resolution, and more likely have higher prediction accuracy by incorporating more predictors, and some use both spatial and temporal data for building climate-BA models that can explain both the spatial and temporal variability. From the stakeholders' perspective, it is not clear how this study, as presented, will further advance their knowledge and help them to improve fire management or climate adaptation strategies.*

Response: We acknowledge that there are studies that develop statistical climate-BA models for regional fire forecasts. For instance Chen et. (2011) develop a predictive fire model for South America linking sea surface temperatures and fire series over the period 2001–2009 and over a grid of 5 degree. They focus on only one fire season severity, defined as a 9-month period centred at the peak fire month. Fernandes et al. (2011) combine a statistical model with dynamical seasonal forecast of SSTs to predict fires in the Amazon considering data aggregated over a grid of 0.25 degree during the period 2000-2009 and for the July-August-September fire season. Westerling et al. (2002) and Preisler and Westerling (2007) develop a statistical model to forecast BA in western US considering data for the

period 1980-2004 at 1 degree of resolution for 6 months of the US fire season (May through October). Spessa et al. (2015) show that severe fire events in Indonesia can be forecast months in advance using predictions of seasonal rainfall from the ECMWF System 4. They analysed fire and climate data over the period 1997–2010 at 0.5 grid cells for the period from June to November. Compared to these, our analysis is innovative in that it estimates BA predictability on a global scale considering each season separately and considering a multi-model database of operational global seasonal predictions. With respect to Fernandes et al. (2011), Chen et al. (2011) and Spessa et al. (2015) our analysis considers more years (which in principle allowing for larger confidence in the analysis results), and, compared to Chen et al. (2011), our analysis is at higher resolution (in the revised version of our manuscript we consider a 2.5 degree resolution).

Our study could serve as the basis for the development of a global fire seasonal forecast product and to identify relevant seasons and regions where the forecast quality of fire predictions is still not well defined. Our approach is not designed to replace the local systems currently in use, but instead it intends to add complementary information to the existing methods. For instance, certain events driving fires (e.g. droughts) usually affect extensive areas beyond national boundaries. Extreme fire seasons can thus affect multiple countries simultaneously, justifying the efforts for a transnational network on fire risk management. A common, global seasonal forecast system could help international resource sharing agreements, thus improving fire management. This comment is now included in the revised manuscript.

4. Referee's Comment: *This paper presented a solid study in terms of testing the forecasting skills of using cumulative preceding SPEI, a drought indicator, to predict area burned at a global scale. It would likely fit a specialty journal better.*

Response: This manuscript illustrates a novel and parsimonious method for seasonal prediction of burned area (BA) at global scale and describes a set of innovative results based on using seasonal climate forecasts to predict BA a few months in advance. This study was designed and performed by a group of scientists with different backgrounds including biogeoscience, climate science, hydrology, physics, and stochastic modelling. The manuscript is an example of a study that tries to bring together the progress in climate modelling to produce the best climate information for assessing climate-driven impacts. Thus, as already commented previously, we think that this study could be of general interest for the diverse readership of this journal as similar approaches could be developed for other climate-sensitive applications.

References

- Chen, Yang, Randerson, James T, Morton, Douglas C, DeFries, Ruth S, Collatz, G James, Kasibhatla, Prasad S, Giglio, Louis, Jin, Yufang, & Marlier, Miriam E. 2011. Forecasting fire season severity in South America using sea surface temperature anomalies. *Science*, 334(6057), 787–791.
- Fernandes, K., Baethgen, W., Bernardes, S., DeFries, R., DeWitt, D.G., Goddard, L., Lavado, W., Lee, D.E., Padoch, C., Pinedo-Vasquez, M. and Uriarte, M. (2011). North Tropical Atlantic influence on western Amazon fire season variability. *Geophysical Research Letters*, 38(12).
- Preisler, H. K., & Westerling, A. L. (2007). Statistical model for forecasting monthly large wildfire events in western United States. *Journal of Applied Meteorology and Climatology*, 46(7), 1020-1030.

- Spessa, AC, Field, RD, Pappenberger, F, Langner, A, Enghart, S, Weber, Ulrich, Stockdale, T, Siegert, F, Kaiser, JW, & Moore, J. 2015. Seasonal forecasting of fire over Kalimantan, Indonesia. *Natural Hazards and Earth System Science*, 15(3), 429–442.
- Westerling, A. L., Gershunov, A., Cayan, D. R., & Barnett, T. P. (2002). Long lead statistical forecasts of area burned in western US wildfires by ecosystem province. *International Journal of Wildland Fire*, 11(4), 257-266.
- Willmott, C. J., & Matsuura, K. (2005). Advantages of the mean absolute error (MAE) over the root mean square error (RMSE) in assessing average model performance. *Climate research*, 30(1), 79-82.

Manuscript: NCOMMS-17-30118-T

OLD TITLE "Predicting climate-driven seasonal variations in wildfire activity worldwide"

NEW TITLE "Predicting climate-driven seasonal variations in fire activity worldwide"

Responses to reviewer #2

Reviewer #2 (Highlight): *The authors present an evaluation of the predictability and skill of global fires based on SPEI as estimated from observations and ECMWF S4 seasonal forecast. Although the topic is current and relevant, I would like to see some issues addressed before the paper is considered for publication in Nature Communications (detailed comments follow).*

Response: We wish to thank the anonymous referee for his/her thoughtful and constructive comments and encouraging opinion. Below we describe how we have addressed each point.

Main comments:

1. Referee's Comment: *The authors argue this study could serve as the basis for the development of a global fire seasonal forecast product. Evaluating the seasonal forecast skill of one GCM is a first step, but I would be more convinced (for the purpose of effectively creating a fire forecast product) if the study was based on multiple forecast models, as ensemble means have typically better skill than any particular model alone (Robertson et al., 2004).*

Robertson, A. W., Lall, U., Zebiak, S. E., & Goddard, L. (2004). Improved combination of multiple atmospheric GCM ensembles for seasonal prediction. Monthly Weather Review, 132(12), 2732-2744.

Response: We fully understand this remark and, according to this comment, we have redone all the analyses considering a larger set of models. We used the systems developed in the EUROpean Seasonal-to-Interannual Prediction (EUROSIP), and the North American Multi-Model Ensemble (NMME). All these systems represent the most comprehensive set to date of seasonal forecasts and reforecasts. It is worth noting that no study so far has attempted the inter-comparison exercise of assessing the relative qualities of seasonal prediction systems to forecast climatic indicators and burned area; an aspect in which this study is pioneering.

2. Referee's Comment: *Line 168: The authors argue that their study serves as a baseline for future analyses, which will benefit from more developments, including a more refined seasonal climate forecast and higher spatial resolution. I suggest the authors revisit this argument as both the seasonal forecast model (ECMWF S4) and observational datasets are currently available at finer spatial resolution than that selected to derive the empirical relationships to BA (5 degrees). In addition, the coarser data set used in this study (GPCP at 2.5 degrees) can be replaced by other precipitation datasets namely CHIRPS, GPCC, CRU that are available at higher spatial resolution.*

Response: We thank the reviewer for this comment. To address this issue, we have performed the analyses again considering a spatial resolution of 2.5 degrees using the GPCP dataset for precipitation. GPCP is one of the most used dataset in climate studies both in general (Sun et al. 2017) and for the specific purpose of validating climate models (Tapiador et al. 2017). In addition, GPCP fully covers our domain and study period, while CHIRPS (that it is not available for areas north of 50°N, where there are also fires) and GPCC (version 7, available up to 2013 only) do not. Analyses such as GPCC and CRU, based only on rain gauge, have several shortcomings, such as missing spatial areal coverage over regions without gauges and representativeness issues since the distribution of stations is highly inhomogeneous, generating severely data-poor regions (e.g. Africa and South America). For more details on this issue see e.g. Mo and Lyon (2015), Sun et al. (2017), Tapiador et al. (2017) and references therein. Finally, although the forecast systems have higher resolution, the compromise resolution needs to be coarser to match the resolution of the observational references.

3. Referee's Comment: *Line 177: The authors conclude that "... globally, the burned area is mainly influenced by concurrent drought and high temperature".*

Mainly in what sense? If the model was averaging, globally, skill of $R=0.62$ in JJA that corresponds to less than 40% of BA variance explained. Please, clarify.

Also, the analysis presented here is based on SPEI and no evidence is presented to show that, for example, a BA model based on precipitation (or temperature) alone could have produced similar results. In other words, the sentence above suggests that among all the predictors, concurrent drought and high temperature are the most relevant, even though no other predictor is used.

Response: According to this comment and in response to a comment from another reviewer, we have redone all the analyses considering different potential predictors. To achieve this goal, we have considered four different model/indicators including: SPEI, as in the previous version of the manuscript; SPI (the standardized precipitation index); temperature (T) alone; and a multi regression model based on temperature and SPI (SPI-T). Results show that models based on SPEI, SPI or SPI-T perform similarly, far better than T. With respect to the sentence in line 177, we meant that the coefficient β , representing the response of BA to SPI, is generally negative (i.e. in most regions drier conditions led to larger BA values). We acknowledge that the sentence was not clear enough. In the revised version, we have clarified this issue by changing the sentence to:

"We show that in most regions the BA is inversely associated with SPI (negative correlation). Given that negative SPI values correspond to dry conditions, this suggests that, as expected, drier conditions lead to larger BA values."

References

- Mo, K. C., & Lyon, B. (2015). Global meteorological drought prediction using the North American multi-model ensemble. *Journal of Hydrometeorology*, 16(3), 1409-1424.
- Sun, Q., Miao, C., Duan, Q., Ashouri, H., Sorooshian, S., & Hsu, K. L. (2018). A Review of Global Precipitation Data Sets: Data Sources, Estimation, and Intercomparisons. *Reviews of Geophysics*.
- Tapiador, F. J., Navarro, A., Levizzani, V., García-Ortega, E., Huffman, G. J., Kidd, C., ... & Roca, R. (2017). Global precipitation measurements for validating climate models. *Atmospheric Research*, 197, 1-20.

Manuscript: NCOMMS-17-30118-T

OLD TITLE "Predicting climate-driven seasonal variations in wildfire activity worldwide"

NEW TITLE "Predicting climate-driven seasonal variations in fire activity worldwide"

Responses to reviewer #3

Reviewer #3 (Highlight): *This study showed the usefulness of seasonal climate forecast in wildfire prediction. Near 50% of the continental areas benefit from significantly skillful forecast of wildfire based on seasonal climate model forecast. The analyses are valid, and physical interpretations of results are provided. The study is informative. I have a few comments here for consideration.*

Response: We wish to thank the anonymous referee for his/her useful and constructive comments. Each specific point has been addressed in the manuscript as explained in the following document.

Comments:

1. Referee's Comment: *The authors described the motivation and aims of the study. Can you also state what the novelty is of this study besides the spatial scale (global scale)?*

Response: In the revised version of the manuscript, we have included more details on the novelty of this study. In summary, the overarching goals of this study are (i) developing empirical predictive relationships between fires and climate variables for the entire globe and (ii) exploring the performance of an integrated climate-BA model that combines empirical fire-climate models with global climate seasonal forecasts, to obtain seasonal predictions of fire activity worldwide. In addition to this explanation, we have added the following paragraph to the main manuscript:

"The key contribution of this study is assessing the current skill of BA predictions using multi-model seasonal climate predictions at a global scale and for each season separately. The results reveal substantial BA predictability based on antecedent and forecasted climate conditions that can be exploited for fire risk management months ahead. Our study can serve as the basis for the development of a global fire seasonal forecast product."

2. Referee's Comment: *When saying DJF and JJA seasons (also SON and MAM), for example in figure 2, do you mean the average of the 3 months in the season? i.e., DJF is the mean of Dec-Jan-Feb? If so, conventionally, I would call the forecasts of JJA initialized in May as lead-1-month forecasts rather than lead-4-month forecasts. Did you use data during 1981-2016 in figure 2?*

Response: When saying DJF, we mean the total burned area for the three months. We changed "with a lead-time of four months" to "with a lead-time of one month". In figure 2 we use data for the period 1995/06-2016/05 as reported in the Section "Methods".

3. Referee's Comment: *To apply this to real wildfire forecasts, you may need calculations for every 3-month moving window, e.g., JFM, FMA, DJF, provided that seasonal climate forecasts are initialized every month. So you will have forecast every month. But for research purpose, this is totally fine.*

Response: We agree and we comment on this possible application in the revised manuscript.

Manuscript: NCOMMS-17-30118-T

OLD TITLE "Predicting climate-driven seasonal variations in wildfire activity worldwide"

NEW TITLE "Predicting climate-driven seasonal variations in fire activity worldwide"

Responses to Editorial Comments

1. Editorial's Comment: *Title: Burned area data does not distinguish whether fires are unplanned (wildfires) or prescribed/controlled fires, thus I suggest the authors explain how they defined wildfires or simply use the term fires.*

Response: We agree: GFED4 data include different types of fire (e.g., wildfires, cropland burning, prescribed fires, etc.). In the revised version of the manuscript, we have changed the term "wildfires" to "fires".

2. Editorial's Comment: *Line 21: Fernandes et al. (2011) combines a statistical model with dynamical model seasonal forecast of SSTs to predict fires in the Amazon. It should be included in citation.*

Response: Included.

3. Editorial's Comment: *Line 54: What is a climatic balance? Do the authors mean water budget?*

Response: We changed "climatic" to "water" balance.

5. Editorial's Comment: *Line 56: Why such a coarse aggregation (5°x5°) is used, given the datasets allow for a more refined grid?*

Response: We redid all the analyses considering a spatial resolution of 2.5 degrees using the GPCP dataset for precipitation. Please see our responses to the comment number 2 of the second reviewer for more details.

6. Editorial's Comment: *Line 89: Since this is a global analysis, I suggest the season names (ex. summer) are replaced by the corresponding trimester.*

Response: Changed.

7. Editorial's Comment: *Line 91: Lead time is defined as the period of time between the issue time of the forecast(May) and the beginning of the forecast period (June for JJA), thus in this case is a 1 month lead time, not 4.*

Response: Corrected.

8. Editorial's Comment: *Line 99: Do negligible ENSO impacts in these regions make the BA forecast more relevant since it is able to overcome that limitation? If that is the case, why is that? Please, elaborate.*

Response: We changed the sentence at line 99 to:

"The regions where significant correlations are found include also extra-tropical areas, such as Mediterranean Europe and central-northern Asian regions, where dynamical forecast systems are known to have a limited prediction skill (Kryjov, 2012; Doblas-Reyes et al., 2013; Madadgar et al., 2016). The skill found here largely relies on

merging observational information (for the months previous to the fire season) with seasonal forecasts (for the fire season)".

9. Editorial's Comment: *Line 160: Sentence: "... the performance of seasonal climate forecast system is still moderate," Add reference. Does this refer to specifically ECMWF S4? Is that true for both precipitation and temperature? Every season?*

Response: We changed the sentence at line 160 to:

"This is especially useful over areas where the performance of the dynamical forecast systems is still affected by significant errors. For instance our models show skill also in mid-latitude regions, where dynamical forecast systems show acceptable skill only for particular seasons and events (see e.g. Frías et al. 2010; Doblas-Reyes et al. 2013; Ceglar et al. 2018)".

10. Editorial's Comment: *Line 227: Sentence: "For this reason, the time series of BA and of the SPEI index are linearly detrended to minimise the influence of slowly changing factors such as gradual increase in fire management effort and land-use changes".*

I would like to point that observed trends in temperature have been shown to influence burned area anomalies regionally (Fernandes et al., 2017). Both observations and GCM hindcasts reproduce regional temperature trends and if the authors wish to place their findings in the context of climate change, as discussed in Lines 178-184, it would be worth to evaluate the sensitivity of BA forecast model to trends.

Fernandes, K., Verchot, L., Baethgen, W., Gutierrez-Velez, V., Pinedo-Vasquez, M., & Martius, C. (2017). Heightened fire probability in Indonesia in non-drought conditions: the effect of increasing temperatures. Environmental Research Letters, 12(5), 054002.

Response: We assess the sensitivity of the BA forecast to trends by running the SPI-BA model with the original (i.e. without detrending) data, obtaining similar results. We now include a comment on this in the revised manuscript and new material as Supporting Information.

11. Editorial's Comment: *Other relevant references (consider citing)*

Di Giuseppe, F., Pappenberger, F., Wetterhall, F., Krzeminski, B., Camia, A., Libertá, G., & San Miguel, J. (2016). The potential predictability of fire danger provided by numerical weather prediction. Journal of Applied Meteorology and Climatology, 55(11), 2469-2491.

Shawki, D., Field, R. D., Tippet, M. K., Saharjo, B. H., Albar, I., Atmoko, D., & Voulgarakis, A. (2017). Long-Lead Prediction of the 2015 Fire and Haze Episode in Indonesia. Geophysical Research Letters, 44(19), 9996.

Response: We added these references.

REVIEWERS' COMMENTS:

Reviewer #1 (Remarks to the Author):

The authors have addressed the reviewers' comments thoroughly for the most part. Here are my comments after reading carefully the revised manuscript and the response file.

1) Abstract: Please mention the predictor (SPI) for the climate-BA models (Line 22).

2) About the model development, which is the key of this manuscript, the authors explored a few other climate variables including SPEI, air temperature, and a regression-based precipitation-temperature indicator.

* "The approach is based on finding the values of the model parameters (β , m and t) that maximize the correlation (r) between modelled and observed BA series". Please clarify what regression or optimization technique was used to estimate the model parameters? For example, the most widely used least square regression, the goal is to minimize the sum of squared residuals between modeled and observed data, which is different from the goal to maximize the correlation here.

* It wasn't clear to me what is the "regression-based precipitation-temperature indicator". Does it mean ($BA = a + b \cdot SPI + c \cdot T$)?

* For the T indicator, what is the rationale for using the standardized series (anomaly divided by standard deviation)? What is the impact on model performance if using the non-standardized T series?

* I think that it would be valuable to add another important variable, Vapor Pressure Deficit (VPD), which accounts for the fact that a combination of high temperature and dry air (low relative humidity) creates the most fire-prone conditions. And there are quite a few studies suggesting VPD is a very important variable for predicting BA.

* Were only univariate models tested and compared, except for SPI-T model? There are quite some statistical approaches available for a more systematic feature selection and model building. Basically you can bring in a set of potential input variables all together, and only the most significant variables will get selected for the final model.

2) I second the suggestion of another reviewer on 3-month moving window approach, which is worth to test in more detail.

Reviewer #2 (Remarks to the Author):

Review of "Predicting climate-driven seasonal variations in fire activity worldwide"

The authors did a superb job at implementing the reviewers' suggestion to the first submission of this work. The topic is relevant and innovative, giving new insights on the potential for combining seasonal forecast from dynamical climate models with statistical techniques to provide seasonal fire forecast worldwide. I recommend its publication provided the authors address some needed clarifications to the method employed.

Main comments:

1) There are some inconsistencies in the explanations of "m" in Fig. 1, the text and Fig. 8.

In line 68. "...m is the month for which the SPI is computed". This sentence can be interpreted as "m" being a calendar month (varying from 1 to 12) and in fact "m" represents a lag with reference to the last month of calculated SPI (varying from 0 to 5). Thus, if my interpretation is correct, SPI3(8) in Fig. 1 should actually be shown as SPI3(0).

2) Merging of OBS and FCST to calculate SPI as depicted in Fig.1

My understanding is that for each gridcell (and season) there will be ONE SPI determined as the best predictor to BA. So, depending on t and m, SPI would be based on OBS or FCST alone. What is not clear to me is how SPI12 is merged. Are the forecast models' precipitation absolute values merged with GPCP observations to derive the precipitation distribution necessary to calculate SPI12? Calculating SPI from OBS and FCST separately makes sense (each will have its own precipitation distribution). However, dynamical models can be biased in their estimate of precipitation amounts and combining these 2 sources may result in a precipitation distribution that is not representative of either OBS or FCST. Could you please clarify?

3) Line 215: "four months ahead"

Technically, is one month ahead as you are relying on 3mo SPI (DJF) that happens to end in February. It would be 4 months ahead if 1mo (Feb) SPI was the predictor to BA and being forecasted in November.

4) Line 241: The sentence: "For applying our approach to continuously-updated fire forecasts, one should resort to seasonal forecasts issued every month for 12 rolling three-month periods (e.g. JFM, FMA, ..., DJF)".

Do the authors mean that for an operational forecast the 4 seasons presented in the paper need to be extended to cover all trimesters of the year? The way I read the sentence at first, it seemed to say that a 12 trimesters forecast is needed every month (at once). Perhaps re-write it for clarification.

5) Line 242: "The development of a prototype real-time operational forecast system, however, may be challenging owing to the uncertainties of the observed near-real time data..."

I would be more optimistic. I suggest the authors check CAMS-OPI dataset. It could provide the tools necessary for a first try at implementing the method (bound to be improved) and provide users with a concrete and actionable product.

(http://www.cpc.ncep.noaa.gov/products/global_precip/html/wpage.cams_opi.html)

6) Line 268: Fire risk is expected to increase where the climate is projected to become warmer and drier.

Not only. Non-linearity and threshold conditions in the relationships between precipitation, temperature and fire risk have been documented.

Fernandes, K., Verchot, L., Baethgen, W., Gutierrez-Velez, V., Pinedo-Vasquez, M. and Martius, C., 2017. Heightened fire probability in Indonesia in non-drought conditions: the effect of increasing temperatures. *Environmental Research Letters*, 12(5), p.054002.

Aldersley, A., Murray, S.J. and Cornell, S.E., 2011. Global and regional analysis of climate and human drivers of wildfire. *Science of the Total Environment*, 409(18), pp.3472-3481.

Point-by-Point Response to Review Comments

The authors would like to thank the Editor, and the anonymous reviewers for the constructive and thoughtful comments and suggestions which led to substantial improvements in the revised version of the manuscript. In the following, the issues raised are addressed point-by-point in the order they are asked.

Responses to reviewer #1

Reviewer #1 (Highlight): *The authors have addressed the reviewers' comments thoroughly for the most part. Here are my comments after reading carefully the revised manuscript and the response file.*

Response: We wish to thank the anonymous referee for the time he/she devoted to review our manuscript.

Referee's Comment: *Abstract: Please mention the predictor (SPI) for the climate-BA models (Line 22).*

Response: Done.

Referee's Comment: *About the model development, which is the key of this manuscript, the authors explored a few other climate variables including SPEI, air temperature, and a regression-based precipitation-temperature indicator.*

"The approach is based on finding the values of the model parameters (β , m and t) that maximize the correlation (r) between modelled and observed BA series". Please clarify what regression or optimization technique was used to estimate the model parameters? For example, the most widely used least square regression, the goal is to minimize the sum of squared residuals between modeled and observed data, which is different from the goal to maximize the correlation here.

Response: The parameter β of the model in Eq. 1 is estimated using a robust regression procedure that adopts iteratively reweighted least squares with a bisquare weighting function (Street et al., 1988). Such an approach is less sensitive to outliers than the classic least-squares estimators. This is explained in the revised text (of tracked-changes version of the paper) in lines 466-469. Once β is estimated for all the tested m and t pairs, we just identify the parameters m and t that provide the highest significant correlation between the modeled and observed BA series. This is explained in the revised version in lines 445-461. The correlation is estimated using the Pearson correlation coefficient; a very common metric used to develop empirical seasonal prediction models and to assess the forecast skill of these system. This is now clarified in the Methods section (line 450).

Referee's Comment: *It wasn't clear to me what is the "regression--based precipitation--temperature indicator". Does it mean ($BA = a + b \cdot SPI + c \cdot T$)?*

Response: Yes, the Reviewer is correct. We have now clarified it in lines 130-131 and 168, where we explicitly describe this model as $BA = \beta \cdot SPI_t(M-m) + \gamma \cdot T_t(M-m) + \varepsilon$

Referee's Comment: *For the T indicator, what is the rationale for using the standardized series (anomaly divided by standard deviation)? What is the impact on model performance if using the non-standardized T series?*

Response: We consider the standardized series for the T indicator to be consistent with the other (standardized) predictors and, as acknowledged in the section Methods, to make the regression results for the grid-points comparable with each other. The model performance does not change using non-standardized or standardized variables, as it does not affect the robust regression fit neither the correlations found.

Referee's Comment: *I think that it would be valuable to add another important variable, Vapor Pressure Deficit (VPD), which accounts for the fact that a combination of high temperature and dry air (low relative humidity) creates the most fire-prone conditions. And there are quite a few studies suggesting VPD is a very important variable for predicting BA.*

Response: VPD is certainly an important fire-related climatic variables, as shown by Seager et al. (2015) for southwestern United State and by Sedano and Randerson (2014) for Alaska. However a global analysis considering this variable (considering the VPD-BA relationship and/or the VPD seasonal predictability) is still missing, also due to data limitations (we do not have a spatially consistent global VPD records). In addition, since VPD is a proxy of the fuel moisture content (see e.g. Williams et al., 2015), it is probably a good fire risk indicator only in regions with abundant fuel, but rarely dry ecosystems, where fires are mainly limited by fuel moisture. However, in regions others than that, VPD could be a less satisfactory predictor (to be explored, but out of the scope of the present study), thus limiting its applicability on a global scale.

To address the Reviewer's comment, we mentioned VPD when we refer to future developments (lines 344-346) in the revised manuscript:

"[...] other climatic variables (see e.g. Williams et al. 2015 that consider the Vapor Pressure Deficit, or Turco et al. 2017c that consider the Standardized Soil Moisture Index [...])."

Referee's Comment: *Were only univariate models tested and compared, except for SPI-T model? There are quite some statistical approaches available for a more systematic feature selection and model building. Basically you can bring in a set of potential input variables all together, and only the most significant variables will get selected for the final model.*

Response: We agree. In fact, we followed such an approach. Our potential input variables are T, SPI and SPEI. Apart from the univariate models, we can consider the SPI-T, SPI-SPEI, T-SPEI and T-SPI-SPEI models. Among these, the only that makes sense is the first one (using a linear combination of SPI and T). The others lack of sense because of the high correlation between SPI and SPEI. In any case, our results show that the univariate models based on SPI produce the most skilful out-of-sample predictions of the impact of climate variability on BA globally.

Referee's Comment: *I second the suggestion of another reviewer on 3-month moving window approach, which is worth to test in more detail.*

Response: We also agree with this suggestion. As we obtain robust signals for all seasons along the year, future developments of the proposed method could consider moving windows to provide "real wildfire forecasts" operationally. We commented on that, along with the limitations and challenges for such an operational implementation of the method in the revised version of the manuscript.

References

- Seager, R., Hooks, A., Williams, A. P., Cook, B., Nakamura, J., & Henderson, N. (2015). Climatology, variability, and trends in the US vapor pressure deficit, an important fire-related meteorological quantity. *Journal of Applied Meteorology and Climatology*, 54(6), 1121-1141.
- Sedano, F., & Randerson, J. T. (2014). Multi-scale influence of vapor pressure deficit on fire ignition and spread in boreal forest ecosystems.
- Turco, M., Levin, N., Tessler, N., & Saaroni, H. (2017c). Recent changes and relations among drought, vegetation and wildfires in the Eastern Mediterranean: The case of Israel. *Global and Planetary Change*, 151, 28-35.
- Williams, A. P., Seager, R., Macalady, A. K., Berkelhammer, M., Crimmins, M. A., Swetnam, T. W., ... & Hryniw, N. (2015). Correlations between components of the water balance and burned area reveal new insights for predicting forest fire area in the southwest United States. *International Journal of Wildland Fire*, 24(1), 14-26.

Responses to reviewer #2

Reviewer #2 (Highlight): *The authors did a superb job at implementing the reviewers' suggestion to the first submission of this work. The topic is relevant and innovative, giving new insights on the potential for combining seasonal forecast from dynamical climate models with statistical techniques to provide seasonal fire forecast worldwide. I recommend its publication provided the authors address some needed clarifications to the method employed.*

Response: We wish to thank the anonymous referee for his/her thoughtful and constructive comments and encouraging opinion. We have done our best to improve the manuscript following the reviewer's recommendations.

Referee's Comment: *There are some inconsistencies in the explanations of "m" in Fig. 1, the text and Fig. 8.*

In line 68. "...m is the month for which the SPI is computed". This sentence can be interpreted as "m" being a calendar month (varying from 1 to 12) and in fact "m" represents a lag with reference to the last month of calculated SPI (varying from 0 to 5). Thus, if my interpretation is correct, SPI3(8) in Fig. 1 should actually be shown as SPI3(0).

Response: The reviewer is right. According to this comment, we redefined the model of Eq. 1 as $BA = \beta \cdot SPI_t(M-m) + \varepsilon$.

The quantity $M-m$ is the calendar month for which the SPI is computed, thus now the Fig. 1 correctly indicates SPI3(8) as the SPI calculated in August. Accordingly, we also changed the caption of Fig. 8 indicating that we show the values of m .

Referee's Comment: *Merging of OBS and FCST to calculate SPI as depicted in Fig. 1*

My understanding is that for each gridcell (and season) there will be ONE SPI determined as the best predictor to BA. So, depending on t and m , SPI would be based on OBS or FCST alone. What is not clear to me is how SPI12 is merged. Are the forecast models' precipitation absolute values merged with GPCP observations to derive the precipitation distribution necessary to calculate SPI12? Calculating SPI from OBS and FCST separately makes sense (each will have its own precipitation distribution). However, dynamical models can be biased in their estimate of precipitation amounts and combining these 2 sources may result in a precipitation distribution that is not representative of either OBS or FCST. Could you please clarify?

Response: Exactly, for each grid point there is one SPI (with specific t and m values) as the best predictor to BA. In some cases, these t and m values make it possible to use observations to compute the SPI (e.g. when $t=6$ or 12, or when $t=3$ and $m \geq 2$). In these cases, we merged the bias-corrected model forecasts with observations. The bias correction, which is based on a simple linear scaling method, is performed to avoid the possible inconsistencies mentioned by the reviewer. We have now included in the Methods section what follows (lines 401-420):

"Specifically, we bias corrected the model ensemble mean at each grid-point considering, for precipitation, a scaling factor based on the ratio of long-term monthly means (over the period 1981-2016) of the observed and simulated data, while for temperature, we consider the difference of long-term monthly means of the observed and simulated data to correct the raw

data (see Turco et al. 2017 for more details). This scaling factor is lead-time and starting-date dependent, thus varying for each forecasted month and for each issued forecast. This bias correction aims at avoiding possible inconsistencies between simulated and observed data when both are merged to construct the predictors”.

Referee’s Comment: *Line 215: “four months ahead”*

Technically, is one month ahead as you are relying on 3mo SPI (DJF) that happens to end in February. It would be 4 months ahead if 1mo (Feb) SPI was the predictor to BA and being forecasted in November.

Response: That’s actually the case. We always assess the skill of retrospective forecasts (or re-forecasts) of BA considering a lead-time of one month. For that, in the specific case mentioned by the reviewer, we use the four months ahead predictions of precipitation issued in November to compute the SPI of February, so that’s why we said “four months ahead”. To make it clearer, we changed this line to (lines 290-292):

“In these cases it is necessary to resort to the four-months-ahead predictions of precipitation to compute the SPI”.

Referee’s Comment: *Line 241: The sentence: “For applying our approach to continuously-updated fire forecasts, one should resort to seasonal forecasts issued every month for 12 rolling three--month periods (e.g. JFM, FMA, ..., DJF)”.*

Do the authors mean that for an operational forecast the 4 seasons presented in the paper need to be extended to cover all trimesters of the year? The way I read the sentence at first, it seemed to say that a 12 trimesters forecast is needed every month (at once). Perhaps re-write it for clarification.

Response: To make this sentence clearer, we changed it to (lines 315-327):

“For applying our approach to continuously-updated fire forecasts to cover all trimesters of the year, one should resort to seasonal forecasts issued every month for rolling three-month periods (e.g. JFM, FMA, ..., DJF).”

Referee’s Comment: *Line 242: “The development of a prototype real--time operational forecast system, however, may be challenging owing to the uncertainties of the observed near--real time data....”*

I would be more optimistic. I suggest the authors check CAMS-OPI dataset. It could provide the tools necessary for a first try at implementing the method (bound to be improved) and provide users with a concrete and actionable product.

(http://www.cpc.ncep.noaa.gov/products/global_precip/html/wpage.cams_opi.html)

Response: Mo and Lyon (2015) compare the observed SPI with GPCP data and with datasets available in near-real time (including the CAMS--OPI dataset) and show large differences over data-poor regions such as Africa and South America. Thus, taking into account these results and the reviewer’s comment, we modified the text as follows (lines 327-333):

“The development of a prototype real-time operational forecast system, however, may be challenging owing to the uncertainties of the observed near-real time data, especially over data-poor regions such as Africa and South America (AghaKouchak and Nakhjiri, 2012; Mo and Lyon, 2015). Thus, although actionable near-real datasets are available (see e.g. Janowiak and Xie 1999; Chen et al. 2002), it is recommended that, before implementing our approach for real-time application, a careful assessment of the available data sets is performed.”

Referee's Comment: *Line 268: Fire risk is expected to increase where the climate is projected to become warmer and drier.*

Not only. Non--linearity and threshold conditions in the relationships between precipitation, temperature and fire risk have been documented.

Fernandes, K., Verchot, L., Baethgen, W., Gutierrez-Velez, V., Pinedo-Vasquez, M. and Martius, C., 2017. Heightened fire probability in Indonesia in non--drought conditions: the effect of increasing temperatures. Environmental Research Letters, 12(5), p.054002.

Aldersley, A., Murray, S.J. and Cornell, S.E., 2011. Global and regional analysis of climate and human drivers of wildfire. Science of the Total Environment, 409(18), pp.3472-3481.

Response: Taking into account this comment, we added what follows (lines 355-356):

“In a changing climate, several possible pathways of fire response can be identified – depending on the expected changes in precipitation, temperature, vegetation and human activities (Hessl, 2011; Aldersley et al. 2011; Fernandes et al. 2017).”